Original research

# Experiences of individuals with rheumatoid arthritis interacting with health care and the use of a digital self-care application: a qualitative interview study

Jennifer Viberg Johansson ,[1] Hanna Blyckert,[2] Karin Schölin Bywall[1,3]

¹Centre for Research Ethics & Bioethics, Uppsala University Department of Public Health and Caring Sciences, Uppsala, Sweden
²Elsa Science, Stockholm, Sweden
³Division of Health and Welfare Technology, School of Health, Care and Social Welfare, Västerås, Sweden

**Correspondence to**
Jennifer Viberg Johansson;
jennifer.viberg-johansson@crb.uu.se

## ABSTRACT

**Objectives** Over the last few decades, there have been significant improvements in the treatment of rheumatoid arthritis (RA), with the development of new treatments and guidelines for teamwork and patient self-care and access to digital tools. This study aimed to explore the experiences of individuals with RA interacting with healthcare. It also looked at how a self-care application, an educational programme called the 'healthcare encounter', improved patient–doctor communication.

**Design** Semistructured interviews were conducted, and qualitative content analysis was performed.

**Setting** The potential participants, individuals with established, or under investigation for, RA diagnosis at rheumatology clinics in Sweden, were asked to participate in the study via a digital self-care application called the Elsa Science Self-care app.

**Participants** Ten interviews were performed with participants from nine clinics following a meeting with the rheumatologist or other healthcare personnel between September 2022 and October 2022. Phrases, sentences or paragraphs referring to experiences from healthcare meetings and opinions about the digital programme were identified and coded. Codes that reflected similar concepts were grouped; subcategories were formulated, and categories were connected to their experiences and opinions.

**Results** Among our participants, three main categories emerged: the availability of healthcare, individual efforts to have a healthier life and personal interaction with healthcare. Participants described that the 'healthcare encounter' educational programme can be a source of information, which confirms, supports and creates a sense of control.

**Conclusion** The participants valued being seen and taking part in a dialogue when they had prepared themselves (observed symptoms over time and prepared questions). The implementation of digital self-care applications might need to be incorporated into the healthcare setting, so that both the patients and the healthcare personnel have a shared understanding. Collaboration is essential in this context.

## STRENGTHS AND LIMITATIONS OF THIS STUDY

⇒ The study employed semistructured interviews to delve into patients' perspectives on interacting with healthcare, allowing for a comprehensive exploration of their experiences.

⇒ An in-depth investigation was conducted of patients' opinions on the impact of a self-care application on patient–doctor communication, shedding light on the effectiveness and potential improvements in this crucial aspect of healthcare.

⇒ The study findings may not be transferable to other disease groups, as the study investigated how this particular group at this disease stage experienced healthcare.

⇒ Another limitation of the study was that only participants who used the app were invited to participate.

## INTRODUCTION

Rheumatoid arthritis (RA) is a chronic autoimmune disease, causing joint pain and inflammation.[1] It affects about 0.5%–1% of the population,[2 3] commonly appearing in joints such as the hands, knees and ankles, often symmetrically. RA can start gradually over weeks or suddenly overnight. Early symptoms include fatigue, malaise and swelling in the small joints, often with morning stiffness lasting over an hour. Moreover, it can lead to complications in various organs.[4 5]

There has been significant progress in RA treatment in recent decades, including new medications, teamwork guidelines, self-care promotion and digital tool accessibility.[6] While there is no cure, medications can relieve symptoms and preserve joint function. Combining treatments for optimal symptom relief often requires adjustments due to new symptoms or side effects. RA care should include patient education and effective communication to empower patients and promote shared decision-making.[7]

In addition to the physical challenges of RA, individuals also experience psychological struggles, such as distress and helplessness.[7] They constantly feel unwell, which can restrict their engagement in daily life and social activities. Moreover, psychological well-being, including self-esteem, body image and relationships, is negatively affected by joint tenderness in individuals with RA.[8] Besides finding the right medication use and hindering the development of disabilities in the joints, guidelines recommend providing education and tools to handle the psychological aspects of the disease.[4] There are recommendations to help patients identify factors and means to manage symptoms such as pain, fatigue, depression, anxiety and sleep problems.[9] Earlier research has emphasised the need for education in self-management skills and counselling to enhance empowerment and improve the emotional dimension of care. Findings support the need for team-based rehabilitation interventions to enhance empowerment in patients with RA.[10 11] Moreover, results from a review of RA patients' self-management support needs to show that individuals with RA have informational, emotional, social and practical needs. Informational needs can be about how to deal with pain, exercise and medication. Emotional support can be from their relatives or healthcare professionals.[12]

Using mobile technology as a complement in healthcare has increased during the last decade and is now a natural part of healthcare-related services in general,[13 14] as well as for individuals with RA who require daily self-management of the disease.[6] The adoption of digital tools has transformed healthcare, offering RA patients access to reliable information, self-tracking and self-management outside the hospital setting. These tools empower patients to prepare for healthcare interactions. Additionally, they have proven to be more effective and cost-efficient in treating RA.[15] When comparing traditional care with collaborative care, it is evident that there is a need for a shared understanding between RA patients and healthcare professionals.

Both approaches emphasise patients' responsibility for their health, but healthcare providers play a vital role in communicating, motivating and aiding patients in self-management.[16] Medical evidence, guidelines and digital technologies are in place for quality RA care. The question is whether Sweden's healthcare system is prepared for more active patient involvement and effective interactions between RA patients and rheumatologists. There is a need to investigate RA patients' experiences of healthcare interactions and the use of digital tools for self-care and communication.

This study aimed to explore the experiences of individuals with RA, or under investigation for RA, interacting with healthcare. It also aimed to determine how helpful a digital self-care application was in improving patient–doctor communication.

## METHODS
### Design
This was a qualitative study based on semistructured interviews.

### Setting and participants
The potential participants, individuals with established, or under investigation for, RA diagnosis (Swedish guidelines for diagnosis), at rheumatology clinics in Sweden, were asked to participate in the study via a digital self-care application called Elsa (https://www.elsa.science/en/; online supplemental file 1). The app was accessible through the Google Play Store for Android and the App Store for iOS. The participants discovered it through various channels, including recommendations from clinic staff or brochures at health centres, digital platforms and social media or through their own search for self-care. Participants were eligible for the survey if they had an RA diagnosis, were aged 18–80 years old and understood and expressed themselves in Swedish. The structure of the study was to first participate in a survey, then perform a specific programme in the Elsa app called the 'healthcare encounter' and, in conclusion, complete a final survey. At the time of the final survey, they could choose to sign up for a follow-up interview.

The specific programme aims to provide basic knowledge to individuals living with a rheumatic disease that can inspire them to make sustainable lifestyle changes and improve their well-being. The 'healthcare encounter' programme focuses on providing basic knowledge about the treatment, medical options, what to expect from meeting healthcare and how the individuals themselves can prepare before the meeting. This educational programme takes about 20 min to complete. Each person could decide for themselves whether they would like to do it all at once or do parts of it over time. It takes about 2–3 min per day to fill in the daily log. See Box 1 for a summary of the content of the educational programme.

## DATA COLLECTION
In total, 18 out of 43 individuals declared their interest in participating in this interview study. The participants were then asked via email to schedule an interview. The study included participants who had appointments scheduled within the near future or within the past month. Overall, 10 interviews were performed after the meeting with the rheumatologist or other healthcare personnel from 9 different clinics in Sweden between September 2022 and October 2022 by the second author (HB).

---

**Box 1   Content of the educational programme**

⇒ Introduction, background and basic knowledge.
⇒ The goal of the encounter (treatment goals).
⇒ The treatment in general (different medication options).
⇒ Long-term collaboration (the healthcare system's role and your role as a patient).
⇒ How to prepare (before the encounter).
⇒ How to act during the encounter.
⇒ How to follow-up after the encounter.
⇒ Checklist.

---

**Table 1** Interview guide used for the semistructured interviews

| Description of the project, the goal and our background knowledge and interest. Can you please tell me about yourself and your life at the moment (warm-up question) |
|---|
| Before your appointment with the healthcare provider—how did you feel before your visit? |
| Probing questions: Did you prepare in any way? If so, how? Was there anything specific you planned to bring up at the meeting? Did you get any help from the program in your preparation? If so, what? Was there any difference in how you prepared now compared to how you usually prepare yourself? |
| During the appointment with the healthcare provider—how was the visit? |
| Probing questions: What did you talk about (feeling in general, treatment, side effects, etc)? What felt good/less good during the meeting? Did you have the opportunity to bring up what you wanted to talk about (treatment options, any side effects, other complaints, what is important to you)? Compared to previous visits, how did you experience this? |
| After the appointment with the healthcare provider—do you remember how you felt after the visit? |
| Probing questions: Was the meeting as you wanted it to be? Would you like to change anything if you could, in retrospect? In retrospect, is there anything that feels unclear after the meeting (missing info, unsure of how to do something, proceeding with self-care)? How have you handled the uncertainty? |
| About the programme—what do you think about the programme? |
| Probing questions: Was there something that you liked in particular? Something you liked less or even disliked? Is there something you missed or would like more of? |

The interviews lasted 44–63 min and were conducted in Swedish via Google Meets or telephone (n=1). We began each interview by asking the participants about their experience of having RA, their treatment and health in general. Thereafter, we asked them to describe their experience before, during and after the healthcare meeting. They were also asked to describe their use of the digital programme, called the 'healthcare encounter', regarding whether it was useful to them or if it was missing any information or function. A semistructured interview guide with open-ended questions[17 18] was developed by the authors (table 1). The interview guide was pilot tested among colleagues and an individual with RA to assess the comprehension of the questions. The sequence of questions was reorganised to align with the temporal aspects of preparations and experiences before, during, and after the medical appointment and the use of the app.

## Analysis

The recorded interviews were transcribed verbatim by a professional transcription company. Thereafter, we listened to the recordings in their entirety to verify the transcriptions. After all the transcripts were read again, meaning units (phrases, sentences or paragraphs), with experience about healthcare meetings and opinions about the digital programme, were identified for further scrutiny. The material contained 344 meaning units in total. Atlas.ti Web[19] and Microsoft Excel (2016) were used to assist in the data management and analysis process. In the next stage of the process, we continued with comparisons of the meaning units, examining their similarities and differences from the perspective of experiences and healthcare options and the digital programme. Open coding of each meaning unit was added, which summed up what was being said in the text. Two of the interviews were coded simultaneously by both authors (JVJ and HB), who jointly discussed what meaning units to identify as well as interpretations and formulations of the codes,

establishing the initial coding framework. The remaining interviews were coded by JVJ. However, when new codes emerged, digital meetings were held with all coauthors to discuss the integration into the existing scheme. Codes that reflected a similar concept were grouped; subcategories were formulated and categories were identified[20 21] by JVJ and thereafter discussed thoroughly with HB and KSB (table 2). Additionally, two meetings were held with the greater project's research group (Personalized medicine in RA by combining genomics, biomarkers, clinical and patient-derived data from the Nordic countries (NORA)) to solicit feedback on the drafted results of both the main categories and the subcategories, accompanied by illustrative quotations. Thematic saturation was reached regarding the aim of the data collection. The authors, JVJ, HB and KSB, bring several years of experience in conducting interviews and analysing qualitative data. Field notes were taken both during the interviews and throughout the analysis process.

## Patient and public involvement

None.

## RESULTS

In total, 10 interviews (7 women, 3 men) were conducted with persons with, or under investigation for, RA (n=1 under investigation with symptoms 2 years back in time and then later, after the interview, the person received a confirmed diagnosis of RA). Disease duration ranged from 1 to 3 years, with one exception, where the person received a diagnosis 12 years ago. The participants were aged 45–76 years old (mean age 56.7) and were from different demographic locations in Sweden. All of them were taking medication, and they were recruited from across the entire country. Moreover, all participants were accustomed to using smartphones and mobile applications in their daily lives. They all described their

**Table 2** Example of the analytical process of the experiences when interacting with the healthcare providers

| Meaning unit | Initial coding | Subcategory | Category |
|---|---|---|---|
| But like [trying to get in touch], you just feel, what the hell, I have no power. | Does not reach the healthcare providers | Accessibility and confirmation of care | Availability of healthcare |
| I have expressed several times to them on the phone, I feel alone. I feel very alone. I think it is hard actually. But then, it's not like I'm lying down and crying about it, but I feel left out. | Feeling alone and left out | To be taken care of | Availability of healthcare |
| He is responsive, he listens, understanding[…] it is a pleasure to go and see him. I never feel any stress or pressure when I talk to him. He is very responsive and listens… he always takes what I say seriously. | He is responsive, listens and explains; he takes what I say seriously. | To be seen and met with interest | Personal interaction |

experience of being diagnosed with RA as a challenge. Those who were still unsure of what their symptoms indicated expressed frustration about not knowing or understanding their body's signals. They had tried various healthcare services that might be helpful and also tried various medications that might work. Having problems linked to other diseases was also described as a challenge. Consequently, they described having to choose which health issue to alleviate first. Accepting the disease was experienced as both a challenge and a necessity. What was also described as a challenge, and a necessity, was finding a balance in life between activity and recovery, work and leisure and physical activity and rest.

The next section describes the participants' experiences of interacting with healthcare providers and the use of the programme, divided into three main categories. Thereafter, nine subcategories were used to classify the discussions. The categories were: (1) availability of healthcare, (2) individual effort and (3) personal interaction. An overview of the categories and subcategories is presented in table 3. Below, these categories and subcategories are described and illustrated using quotes.

**Table 3** Categories and subcategories of participants' experiences when interacting with healthcare providers

| Category | Subcategory |
|---|---|
| Availability of healthcare | To feel prioritised |
| | Accessibility and confirmation of care |
| | To be taken care of |
| Individual effort | Taking care of myself |
| | Opportunity to equip myself |
| Personal interaction | To be seen and met with interest |
| | Gain self-efficacy |
| | Have a dialogue |
| | Met with competence |

## Availability of healthcare

The category *availability of healthcare* was expressed in three different ways: *to feel prioritised, accessibility and confirmation of care* and *to be taken care of.*

### To feel prioritised

This subcategory focuses on the perception of time availability for patients and healthcare personnel to discuss symptoms, treatment options and resource allocation. About half of the participants felt prioritised and supported by their healthcare team, receiving excellent care. However, the other half reported feeling neglected during their interactions with healthcare providers. They found care meetings to be too short, lacking sufficient time to discuss matters related to their disease beyond medication, and felt unable to ask follow-up questions. The time constraint hindered the establishment of a personal connection. Some even felt dehumanised, that is, treated like they were part of an assembly line rather than as individuals.

> Well, I would have liked his [the rheumatologist] clinical part first[…] go through the joints and any blood tests and explain what the result means, so that you might learn something yourself. So you don't have to keep asking more times. And I wish him to ask, 'How does everyday life work for you?' I still have not been asked that question since 2019. (Respondent 3)

Some participants suggested the benefit of having more frequent meetings early on in the process, especially when more issues arise and there are concerns. The possibility of having additional meetings, if necessary, gives the impression that you are valued. Meeting healthcare personnel who are unfamiliar with rheumatic disease can lead to feelings of insecurity, creating a perception that this patient group is not given priority.

…then I felt some insecurity. If it were someone who had the actual expertise, then they may have asked other follow-up questions. The person was very nice, but it felt unsafe. It felt like, who is taking care of me? It's not her fault, of course, but that's how I felt… (Respondent 7)

### Accessibility and confirmation of care

Having healthcare that is accessible is greatly appreciated and valuable for the participants. This subcategory focuses on access, contact and knowing where to seek assistance. Some expressed that it was easy to make contact via phone or messages via the digital platform used in Sweden (named 1177). When problems arise with symptoms or side effects, fast response, action and reply are appreciated, giving a calming feeling. Having regular meetings scheduled also led to feeling safe.

I think it [meeting the rheumatologist] feels safe. I think it feels great. So, I'm happy to continue with that. (Respondent 5)

However, not all participants experienced this. Most found it difficult to access healthcare through phone calls or messages. One respondent described it as a 'whispering game,' where they would call and relay the problem to a healthcare personnel, who would then forward the message to the rheumatologist. The rheumatologist would then send a message back through a third person. These challenges in reaching healthcare services are perceived as unsafe care, leading to a feeling of being exposed to danger.

…and I know that healthcare is generally heavily burdened, so without talking badly about an individual. But like, you just feel, what the hell, I have no power. (Respondent 7)

The fact that the healthcare services provide a confirmation is also appreciated and valuable to the participants. Some described that the rheumatology clinic does a fantastic job and that you, as a patient, get quick answers if you have concerns or experience practical obstacles with medications. They also stated they receive a good follow-up of blood test results. However, for other participants, it can take a long time to get confirmation of test results. Sometimes it is not until the next appointment or if they have indicated they are not doing well.

The participants who experienced collaboration regarding the treatment of RA expressed that the various healthcare professions had good accessibility and followed up on their work, and it was clear how to reach out to them when a problem occurred.

I have to say that I am very impressed with the rheumatologist at the hospital […] and I do that [send a message] and it can take within an hour then I've got a response that either the nurse has answered, or she says that 'I've passed it on to your doctor' and then in the afternoon, I get an answer. So, this is how it has been for these four years, fantastic. (Respondent 5)

### To be taken care of

Participants in the study appreciated healthcare personnel who made them feel taken care of; they expressed gratitude for the teamwork they witnessed. Moreover, they found that receiving the right support gave them a sense of trust and reassurance during the difficult times. The use of a digital self-care application was seen as affirming their experiences and creating a feeling of being seen. Participants were positive about receiving answers to their questions and concerns via email or chat, as it allowed them to digest new information at their own pace. However, one respondent felt that the application's content did not reflect the reality of healthcare and described it as 'too good to be true'.

Other individuals' stories in the app, […] I just… Yeah, but damn. That's me! (Respondent 4)

A noble example, shared by one of the participants, was when the healthcare professional helped to refer her onward, instead of letting the patient seek help again on her own. Some of the participants experienced great teamwork and expressed pride in their care unit, stating that the staff had a nice and welcoming attitude, and the feeling of being looked after was strong.

…then he [the rheumatologist] had been talking to the gastroenterologist and discussed with them and put together an action plan with them. (Respondent 6)

Some participants expressed frustration and insecurity due to a lack of follow-up and communication from the healthcare services regarding their symptoms, tests and medication. Not hearing from healthcare services made them feel that they were not being cared for. Some participants believed that they had to time their appointments to be sick enough to receive the necessary help and felt that their symptoms were not taken seriously. Others struggled to find the right help and experienced difficulties in navigating the healthcare system, with different units not communicating with each other. Some participants had to take on the role of coordinating their own care, which was tiring and challenging. Finding the right care and prioritising symptoms required significant effort and energy. In complex situations, such as having multiple symptoms and other diseases, participants struggled to find the appropriate care. Some considered changing to a larger hospital for better care. Overall, there was a lack of clarity in knowing where to seek help within the healthcare system, resulting in frustration and being sent around to different units.

You are sent around a lot between the different healthcare units, depending on what problems arise. And there is no connecting link. One part of

healthcare does not know what the other part is doing. (Respondent 6)

## Individual effort

The category of *individual effort* was expressed in two different ways: *taking care of myself* and having the *opportunity to equip myself.*

### Taking care of myself

All participants expressed that they have tried to adapt to life in different ways, describing different attempts at self-care: adjusting their diet, trying different forms of exercise and training and trying to find a balance between rest and activity. The participants described how they prepared themselves in different ways before the appointment with the rheumatologist. In addition to taking notes on how they feel—either with a pen and paper or via the digital tool—they sought information in different ways in order to equip themselves with knowledge. Some stated that equipping themselves with new knowledge was for their own sake. Others stated that it was also to facilitate healthcare, so they can be involved in the 'detective work' of finding effective treatment. Some expressed frustration that they must know a lot themselves to get help, that you must stand up for yourself to get good help and that it feels like one has to be healthy to be able to be sick.

> I don't think it's okay. I told her […] all the side effects. Then it was like 'Well, okay, let's try something else.' I have had to argue a lot to change my medication… (Respondent 5)

Moreover, they try to understand and interpret their test results on the health and medical care's digital platform (name 1177) before the meeting. Many participants appreciated the application tool as a trustworthy source of information with a good spirit.

> So, I google… being critical of sources, it's not that easy as a consumer. This [the application] feels very serious and here, they have talked to individuals who [are knowledgeable and experienced] … that's the image I have, anyway. (Respondent 7)

They valued being able to learn more about: what to expect with the disease, the medication, responsibilities of the different healthcare personnel, how to inform relatives and what activities to engage in to feel better. Learning from others' experiences was also appreciated. One individual expressed that the self-care app could be useful even if one was not yet diagnosed.

Writing down questions, symptoms and how they have felt over a longer period was a common action. For some, this was useful as a tool to reflect on their health, understand the reasons for the symptoms and how they come and go, but also to remember the ups and downs of their symptoms.

> I had a long list I had written before the phone meeting, where I tried to think 'How long have I been in

pain?' 'Where do I have pain?' 'How do I react?' and 'What medicines have I taken?' and so on. So that I don't forget anything when I talk to the physician. (Respondent 6)

The participants thought the self-care app was fun, helpful, and a way to take care of oneself when filling in the daily log. For some, it gave them time to reflect on how they feel and how the symptoms have changed over time. Some thought it would be helpful to keep a log on long-term pain, diet, physical activity, general mood and sleeping pattern. They also expressed a need to be more specific in describing the exact location of the pain (eg, where in the hand). Moreover, they suggested making a note of the possible reason for the pain by writing down the activity they were performing before the pain occurred. On the other hand, some participants said they forgot or felt they did not want to log their symptoms when everything was fine.

> I usually go in and use it [logging symptoms], although it's become a bit more sporadic now the last… well, last month and so. Because I have felt better and then when I feel better, I kind of forget to fill it in. (Respondent 9)

### Opportunity to equip myself

There was a great variation in the extent to which our participants had the opportunity to be educated by their rheumatology clinic regarding their disease. Some had 4 days of education, some had 1 day or even half a day and some did not receive any education. To receive education about the disease, medication, exercise, lifestyle habits and how to find a balance in life was much appreciated by the participants who experienced these aspects at their clinic. They felt it made them feel safe and satisfied, even though life had changed with the diagnosis. Education was perceived to empower them and made it possible to take care of themselves.

> …the introductory education… I mean, four full days. And so, they have it regularly […] with everyone who is newly diagnosed, where they go through [everything]. (Respondent 6)

Those who did not receive education from the rheumatology clinic expressed a desire for it and hoped and even expected the healthcare services to provide it. Some described with surprise and disappointment that they had not received it. Understanding one's illness is deemed important; one wants to understand and take power over one's life. Specific aspects for which they wanted more knowledge were medical treatment, side effects, dietary suggestions, diagnosis and what to expect from life.

> …training and such, they talk about that and what is good. That you shouldn't smoke and like… yes, and so on. But diet, in particular, seems to be taboo. (Respondent 8)

The digital self-care application was perceived as a good tool to help prepare before the meeting with the healthcare providers. Specifically, it increased their self-esteem, helping them to believe in themselves. After going through the programme, participants created a list of good things to prepare prior to the healthcare meetings to be helpful. The list helped them to focus on the most important parts and not forget what is important. Looking back on their personal log was also appreciated as a memory support tool. Some participants suggested that the digital self-care application should promote healthy lifestyle choices and include personal goals and interim goals. The inclusion of information and educational material for loved ones or acquaintances was seen as positive, as it can ease the burden and foster good relations among individuals' support network. The function of logging symptoms was seen as a good basis for a discussion with the rheumatologist. Some participants wished that the healthcare services had access to their logged in data to improve conversations and treatment options. However, some felt that constantly being reminded of their illness through the log can be burdensome.

> …there is a risk that going on and on like that [with reading and log symptoms every day]… that you dig into your illnesses, and I don't think that feels very good. (Respondent 2)

### Personal interaction

Finally, our analysis revealed experiences related to personal interaction, and how the healthcare professionals' attitudes made the participants feel. This is the most comprehensive category regarding the participants' narratives. This category was expressed in four different ways: *to be seen* and *met with interest, have a dialogue, gain self-efficacy*, and *met with competence*.

### To be seen and met with interest

There was a strong emphasis on the appreciation of being seen, namely that professionals would show empathy towards the patient. The participants focused on the personal interaction with the physician and whether one feels that he or she is being attended to individually. They wanted to feel as though they were believed and to gain some hope in difficult situations and not perceived as they were whining. Moreover, they wanted to be seen as a whole individual and talked to as an equal and not have the personnel look down on them. When the doctor takes the time and listens, the feeling of being seen as a person increases.

> Very professional, very calm, and nice and she gave me hope. (Respondent 5)

Some expressed feeling frustrated, stupid and disregarded by rheumatologists who did not listen to their concerns or consider their perspective. They believed that a narrow focus on medication and side effects, without addressing other aspects of the disease, was unhelpful. Patients valued rheumatologists who took the time to listen, showed professionalism and respected their needs. However, if healthcare professionals did not fulfil their promises or pay attention to their medical records, patients felt disregarded. They appreciated rheumatologists, who were knowledgeable and prioritised the need to discuss relevant topics during the appointment. They believed that a collaborative approach, where both the doctor and the patient shared their expertise, improved the encounter. Patients felt more comfortable asking important questions when the rheumatologist took their time and listened to them. Some participants described feeling like just a body to be examined by the rheumatologists and felt unable to discuss their specific pain and symptoms. One participant felt that her medical history was uninteresting to the rheumatologist, while another admitted to lying about her pain levels to ensure her complaints were taken seriously. Patients wanted the physician to consider their lived experience of symptoms and side effects during the treatment discussions.

> … they don't know me. I've met him once, and yet, they somehow assume that… you need to bite the bullet. It's a bit like that. And the first time ever when I was at the health center, when my fingers hurt… and then I had red knuckles. They were really inflamed. Then the medical center doctor told me that I have to learn to live with that; you just have to bite the bullet. (Respondent 8)

### Have a dialogue

The participants appreciated having a good dialogue with the rheumatologist; being able to reason with the rheumatologist about medical treatments and side effects and finding a balance between things in life were perceived as valuable. Moreover, the participants appreciated the rheumatologist taking advantage of the preparation done by the patient before the meeting. Being able to reason about the amount of medication was also appreciated.

Those who did not have the opportunity to have a good dialogue expressed sadness or frustration about what was missing for them. They all described they had heard that other patients had received better care. Lack of a good dialogue caused the participants to feel they could not trust the rheumatologist; thus, they either wanted to change doctors or choose their own paths when it came to medication or lifestyle changes. For some, it was not clear whether it was ok to be critical or even change their rheumatologist if the communication did not work out well; there was a fear that it might result in bad consequences.

> So, actually, I would like to change my physician. Because there is another doctor who helped me at some point when the joints in my hand were painful. He was very nice, accommodating and calm, and such. So actually, I want him. But then you don't know if you dare to change to another physician like that, because then […] what if I get the evil eye […].

So, I feel, it's better I go there to those meetings with him, and he does what he has to do, then I google and learn by myself. (Respondent 3)

There was a feeling of resignation among those who had prepared beforehand but were not listened to. Even though they prepared well (what to ask and written down how they have felt during the past month), the rheumatologist did not take notice of it. They shared that the doctor did not even look them in the eyes, was not interested in the patient's notes or questions and answered using difficult language. Instead, the rheumatologist proceeded with a clinical test and made some comments about the medication. These participants felt they were being ignored, while they could instead have been a valuable resource.

[…] even though I had kept a log for a very long time, there was nothing that he [the rheumatologist] cared about when I was there. So, he looked there and then, the day I was there: 'Are you in pain here, yes. Are you not in pain here, no'? And then it was fine with that. (Respondent 8)

### Gain self-efficacy
Something that was expressed by all participants was the need for encouragement and hope for the future. Healthcare was seen as an important component in their pursuit of a life that was functional and of quality. It is important to be able to express yourself and be taken seriously; then, there is a feeling of being competent in dealing with the new situation. For some, seeking healthcare services can often create a feeling that one has become old and soon dying. Getting help to set new goals and subgoals was perceived as helpful; it helps to believe in the future.

… I have thought about it quite a lot and that helps me, that I need to have a goal. Not a dream, a goal. Everyone has dreams, but then you can pick certain dreams and say that is a dream, and this is my new goal. The goal for me is to get back on the police motorcycle. And being able to function and not have to think about how I'm paddling a kayak with my wrist or how I'm holding the weight or the dumbbell, or whatever it is. But I should just be able to be, as I was before. (Respondent 3)

### Met with competence
If the rheumatologist is knowledgeable and competent, it infuses confidence in the patient. Some participants are impressed by the rheumatologist's skills in balancing the trade-off between deploying different medications at different times. Those participants felt that they received all the answers they needed to remain calm and reassured, and following up on any side effects or symptoms was appreciated.

…questions about the medicine to the rheumatologist… she is good at explaining. (Respondent 4)

Two participants felt they received better care if they were prepared and knew about things themselves. Some participants, however, were disappointed as they did not receive answers to their questions and felt the rheumatologist was only guessing when responding to them, resulting in them feeling resigned.

No, but he seems not to care; it feels so stiff when I talk to him. I had to remind him […]. Well, he's not well-read and that's what annoys me so much when he calls. (Respondent 10)

## DISCUSSION
This study emphasises the importance of patient-centred care for individuals with RA. It highlights the significance of considering patients' psychological well-being, together with their physical health. This is in line with the European Alliance of Associations for Rheumatology recommendations and earlier research.[7 22] The study found that patients valued time and an open dialogue with rheumatologists, and appreciated when their personal experiences of the disease were considered during discussions about symptom relief and treatment options. The study also mentions the psychological challenges faced by individuals with RA, such as distress and helplessness, and the need for acceptance and support from healthcare providers. Collaborative teamwork and clear communication were identified as factors that contributed to patients' overall well-being and disease management. The study recommends that healthcare providers ensure that patients with RA have a comprehensive understanding of their treatment plan, expectations and available support through effective communication and education.

Psychological well-being has been found to have a negative correlation with joint tenderness in individuals with RA.[8] In meetings with rheumatologists, a holistic approach is needed to be able to address issues related to one's well-being. The diagnosis of RA often elicits feelings that are overwhelming, a sense that life has taken an unexpected and unwanted turn, and sadness, anxiety and fear for the future, necessitating professional support.[23] While the immediate focus typically revolves around finding the right drug therapy for inflammation and pain reduction, psychological needs should also be addressed. Some participants in this study expressed frustration at not being seen as individuals, but rather as bodies to be examined, a practice that goes against the recommendations.[22]

Expanding on the importance of self-management, addressing the various needs of individuals with RA—such as information, emotions, social support and practical guidance—is crucial.[24 25] Prior research and guidelines have emphasised the value of education and counselling to empower patients and enhance emotional well-being while providing guidance on managing symptoms such as pain, fatigue, depression, anxiety and sleep issues.[9–12 23 26] Our study highlights that individuals with a disease might feel alone and disappointed due to insufficient attention

to their overall well-being. To address this, healthcare professionals should clarify their roles and discuss expectations with patients to ensure comprehensive support. Additional healthcare contacts focusing on well-being may benefit these patients. An unclear understanding of the physician's role contributes to dissatisfaction, emphasising the need to clarify different healthcare professionals' roles. While participants did not complain about the medical treatment, some experienced negative patient–physician interactions. Minor changes, such as including interested responses about the patient's experience, could improve the quality of these interactions.

Determining the right RA treatment is challenging due to the chronic nature of the disease. Self-management with digital tools offers a promising approach to address RA patients' needs, enhance health outcomes and boost satisfaction with healthcare interactions.[27] Our participants expressed that their well-being varies day-to-day due to symptoms and treatment effects. They bear the responsibility of managing their daily lives, including handling pain, fatigue and medications. Thus, much of the health-promoting activities for RA patients take place outside the rheumatologist's office. Participants valued healthcare personnel considering their observations in disease follow-up. However, some had disappointing experiences. Therefore, we would like to argue that strengthening self-management by using a digital tool to keep a daily log on what happens between visits should be promoted by healthcare, not by the person with RA. From the healthcare perspective, digital tools for self-management may increase the efficiency and time spent during visits, improving relationships with patients and using more precise data to guide treatment decisions.[6]

Various digital tools can enhance access to care in regions lacking specialised teams for this patient group. They can expedite urgent care, boost effectiveness through closer follow-up via web-based education, videoconferencing, telephone or electronic messaging, as successfully seen in the oncology services.[28] It has also increased in healthcare services, in general, during the COVID-19 pandemic.[29] A systematic review assessed whether virtual care options can replace in-person rheumatologist visits while maintaining comparable care quality for RA patients. They found limited evidence on the impact of virtual rheumatology care, with no significant differences in patient outcomes between care provided by rheumatologists and rheumatology nurses. Virtual care offered additional benefits, including improved treatment adherence, functional status and quality of life.[30] However, it is critical that those digital tools are evidence-based and focus on self-management interventions that are developed by healthcare providers and persons with RA to support and empower individuals.[31]

## Strengths and limitations

This study aimed to understand how patients view healthcare interactions and the impact of a self-care application on patient–doctor communication. However, the findings may only be applicable to this specific disease group at their stage of the disease. The study excluded participants who did not use the app, which may limit the generalisability of the results.

There are potential limitations in self-reporting bias and subjectivity in an interview-based study. To address these limitations, the researchers encouraged honest and candid responses, maintained confidentiality, reflected on their own perspectives and biases, sought input from colleagues and used analytical frameworks to improve objectivity in data analysis.

## CONCLUSION

The study emphasises the importance of healthcare providers, including physicians and nurses, addressing areas of patient dissatisfaction. It is important for them to actively listen to the patient's concerns, enhance communication and adopt a patient-centred approach. By addressing these issues, healthcare providers can improve the overall patient experience and satisfaction.

As technology becomes a permanent fixture in our society and healthcare, we must consider how to effectively integrate digital tools. This involves striking a balance between digital and physical meetings, patient preparation and the healthcare service's role in treatment. The gap between patient expectations and healthcare implementation may explain their feelings of being unheard. The next phase should focus on implementing digital tools in healthcare services to establish a shared understanding between patients and healthcare providers. In this particular setting, it is essential to emphasise the pivotal role of collaborative engagement between patients and healthcare providers.

**Acknowledgements** The authors' greatest gratitude goes out to all the participants who participated in the interviews. The authors would like to thank for their time and for sharing their experiences and opinions so generously.

**Contributors** JVJ, HB and KSB conceptualised the study. HB performed the interviews. JVJ and HB analysed the interviews and interpreted the data. JVJ was the major contributor in writing the manuscript. HB and KSB revised the manuscript substantively. All the authors read and approved the final manuscript. JVJ is the guarantor and take full responsibility for the work and the conduct of the study, have access to the data, and controlled the decision to publish.

**Funding** This project was supported by Vinnova, Innovationsfonden and The Research Council of Norway, under the frame of Nordforsk (Grant agreement no. 90825, Project NORA). The funding agreement ensured the authors' independence in designing the study, interpreting the data and writing and publishing the report.

**Competing interests** KSB and JVJ have no conflicts of interest to declare. HB is employed at Elsa Science as a Health educator and User researcher.

**Patient and public involvement** Patients and/or the public were not involved in the design, or conduct, or reporting, or dissemination plans of this research.

**Patient consent for publication** Consent obtained directly from patient(s).

**Ethics approval** This study involves human participants and was approved by the Swedish Ethical Review Authority (Dnr: 2021-05431-01). Participants gave informed consent to participate in the study before taking part.

**Provenance and peer review** Not commissioned; externally peer reviewed.

**Data availability statement** No data are available. The authors have not consented for open access.

**ORCID iD**
Jennifer Viberg Johansson http://orcid.org/0000-0001-9533-9274

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
