## [Reviewer comments · BMJ Open]

ARTICLE DETAILS

TITLE (PROVISIONAL)	Experiences of individuals with rheumatoid arthritis interacting with health care and the use of a digital self-care application: a qualitative interview study
AUTHORS	Viberg Johansson, Jennifer; Blyckert, Hanna; Schölin Bywall,, Karin

VERSION 1 – REVIEW

REVIEWER	De Meyst , Elias KU Leuven, Skeletal Biology and Engineering research center
REVIEW RETURNED	19-Mar-2023

GENERAL COMMENTS	Methods need further explaining: - On which base were patients with RA selected to participate in this study? How did the selection process take place?- It is not really clear why 10 interviews were performed out of 18 patients who were interested. Did they all have no follow-up appointment? Flowchart?- According to which criteria were patients defined to have an established diagnosis of RA?- What do you mean with the one patient "under investigation for RA"? What kind of symptoms does this patient have, is his/her situation reflective of the situation of a patient with RA?- More information regarding the digital self care-care application, e.g. how long are patients supposed to spend time on this app? On a daily basis? And is it free, is it widely available? Study limitations are only very briefly discussed and quite underexposed. English language is adequate. In some sentences "health care professional/provider" would be a more appropriate choice of term, rather than using "health care".
---

REVIEWER	Lwin, Cho University of Medicine, Rheumatology
REVIEW RETURNED	25-May-2023

GENERAL COMMENTS	Thank you for the hard work put on the manuscripts. There are a few things to consider as follow: 1. The link to self-care application was given but it will be more convenient for the readers to give summary information on how it works.2. Page 4
---

	Strength and limitations of the study – did not really described the strength, but only the limitations. 3. Page 5, line 18, a repeat sentence from the abstract. 4. 344 meaning units should be described as an appendix. 5. In the "Health care encounter" the examples of the scripts on "Accessible health care" and "to be seen" are similar in essence. 6. In the subcategory "Healthcare that confirms", the sample scripts revealed mixed description. Please make clear if it is focusing on - conformation of the blood tests or the swiftness of response or clear reply from the healthcare provider. It is confusing. 7. The sample scripts in the the subcategories "To navigate in healthcare" and "to be taken care of" are of essentially the same theme with opposite views from the different patients. 8. The sample scripts in the the subcategories "To be seen" and "Met with interest" are also similar in theme. 9. Discussion section is overloaded with unnecessary information especially on page 21 and 22. 10. Conclusion section needs suggestions on healthcare providers such as doctors and nurses to whom some patients are not satisfied with. 11. The results should display the overall impression on the each subcategories (Eg - most patients perceived that they felt prioritised in the "To feel prioritised"subcategory). 12. Should describe if there is possibility of the bias in their response, if possible. (Eg. new patient who did not reach to a diagnosis yet may feel frustrated) 13. The reference section contains some out of date and irrelevant references. It would be nicer to rewrite especially the results section to relook if some subcategories can be combined, to remove some unnecessary and irrelevant information in the discussion section and add some more recommendations in the conclusion section.
--	--

REVIEWER	Hsiao, Betty Yale University School of Medicine
REVIEW RETURNED	20-Jun-2023

GENERAL COMMENTS	Thank you for this very interesting paper. Johansson et al evaluated the experiences of patients with RA and one patient under investigation for RA with the healthcare system and their input on the use of a digital self-care application via qualitative interviews. Evaluating patient experiences with healthcare is very important and I appreciate the authors' work. I have a few comments about this study: 1) Regarding the patient participants, it may have been more helpful to collect more data on their demographics and report in
--

	table form (eg are they under the care of a rheumatologist? are they taking DMARDs? what are their preferences for using technology? etc) 2) I have concerns that patients with RA and the patient being investigated for RA are lumped together in this study--while there is only one patient being investigated for RA, they may not have RA and therefore should not have been included. Also, would persons under investigation for RA have different answers? Should they be in a different category altogether? 3) I have concerns about the methodology--while authors (JVJ and HB) coded 2 interviews, the rest of the interviews were only coded by 1 person (JVJ). In a qualitative study, multiple coders can help check interpretations against the data, and I would recommend that the remaining 8 interviews are also coded by a second coder and to ensure intercoder reliability, report kappa statistic. 4) My other suggestion is to cut down the length of the paper and to perhaps use a stable format to present the data for clarity (with major and minor themes, with supporting quotes).
--	---

VERSION 1 – AUTHOR RESPONSE

Reviewer: 1	Authors
Methods need further explaining: - On which base were patients with RA selected to participate in this study? How did the selection process take place?	We have described the setting better. “The potential respondents, people with established, or under investigation, RA diagnosis (Swedish guidelines for diagnosis) and being affiliated with a rheumatology clinic, were asked to participate in the study via a digital self-care application called Elsa (https://www.elsa.science/en/). The assessments of the app were from both Google Play Store for Android and Appstore for iOS. They find it out via the clinic (recommendation by staff or brochures at the health centre), via digital platforms and social medial, or by their own search for self-care. Respondents were eligible for the survey if they had an RA diagnosis were aged 18-80 years, and understood and expressed themselves in Swedish. The structure of the study was to first participate in a survey, then perform a specific program in the Elsa app called "The healthcare encounter" and, in conclusion, do a final survey. In the final survey, they had the chance to sign up for a follow-up interview. It was voluntary to sign up.” Page 7, Method, Respondents and the setting
- It is not really clear why 10 interviews were performed out of 18 patients who were	We apologize for the lack of clarity on this matter. Individuals who had their appointments scheduled

interested. Did they all have no follow-up appointment? Flowchart?	in close proximity or had recently had their appointment were included. Additionally, we reached a saturation point with 10 interviews. We have added: “The study included participants who had appointments scheduled within the near future or within the past month.” Page 7, Method, Data collection
- According to which criteria were patients defined to have an established diagnosis of RA?	The potential respondents, people with established, or under investigation, RA diagnosis (Swedish guidelines for diagnosis) and being affiliated with a rheumatology clinic, were asked to participate in the study via a digital self-care application called Elsa. We have described this better now on page 7, Method, Respondents and setting.
- What do you mean with the one patient "under investigation for RA"? What kind of symptoms does this patient have, is his/her situation reflective of the situation of a patient with RA?	The patient mentioned as “under investigation for RA” were someone who had shown symptoms suggestive of rheumatoid arthritis and was currently undergoing medical evaluation to confirm the diagnosis. Her situation reflects a patient who is in the process of receiving a diagnosis for RA. It is not uncommon for the diagnosis to take some time as it typically involves multiple assessments, consultations, and tests. In this case, the patient expressed interest in the meeting and later received confirmation of their condition. We included her due to that, under our analytic process, we got a confirmation that she was diagnosed. Moreover, in this study, our primary focus lies on the patient’s interaction and experience with the healthcare system in the journey of RA, rather than the specific aspects of their RA diagnosis. We are interested in understanding the dynamics of the patient’s journey, including their initial meeting with healthcare professionals, the process of receiving a diagnosis, and the overall experience during this period. The aim is to gain insights into the patient’s perspective and gather valuable information that can contribute to improving the healthcare delivery and support provided to individuals in similar situations. We have added in the article:

	“In total, 10 interviews (seven females, three males) were conducted with persons with RA, or under investigation (n=1 under investigation with symptoms two years back in time and later after the interview that person received a confirmed diagnosis of RA).” Page 10, Results
- More information regarding the digital self care-care application, e.g. how long are patients supposed to spend time on this app? On a daily basis? And is it free, is it widely available?	Regarding the digital self-care application, patients are encouraged to use it on a daily basis or as often as needed. It is recommended to spend a reasonable amount of time on the app for reporting and documenting relevant information to ensure the collection of accurate and valuable data. It is not necessary to spend several hours a day using the app. Typically, patients can spend around 2-3 minutes per day, but they are welcome to spend more time if desired. The application is available for free and can be downloaded in both Swedish and English languages. It is accessible to users in Europe and the United States. We have described this in more detail: “The specific program aims to provide basic knowledge to individuals living with a rheumatic disease that can inspire them to make sustainable lifestyle changes and improve their well-being. Page 7, Method, Respondents and the setting And “The healthcare encounter” program focuses on providing basic knowledge about the treatment, medical options, what to expect from meeting healthcare, and how the individuals can prepare themselves before the meeting. This knowledge program took about 20 minutes to undergo. Each person could decide for them-self whether they would like to do it all at once or split it over time. The daily log can be spent around 2-3 minutes per day. See Box 1 for the content of the knowledge program.”

	Page 7, Method, Respondents and the setting
Study limitations are only very briefly discussed and quite underexposed.	We have provided further details and elaborated on the limitations in our manuscript. “It is important to acknowledge the potential limitations of self-reporting bias and subjectivity in interview-based studies. Self-reporting bias can arise due to participants’ inclination to provide socially desirable responses or selectively recall information. To mitigate this bias, we did our best to use strategies such as emphasizing the importance of honest and candid responses and ensuring confidentiality. Subjectivity is another limitation that needs to be addressed. We were aware of our own interpretations, biases, and preconceived notions that can influence data analysis and reporting. Engaging in reflexivity and maintaining reflexivity journals can help us to reflect on our own perspectives and potential biases. We were seeking input from peers or colleagues to provide valuable insights and alternative viewpoints. We employed an analytical frameworks or coding schemes in reputed dialog to enhance objectivity in data analysis.” Page 26, Strengths and limitations
In some sentences "health care professional/provider" would be a more appropriate choice of term, rather than using "health care". Reviewer: 2	We have reviewed this wording throughout the manuscript. We appreciate your attention to this matter.
1. The link to self-care application was given but it will be more convenient for the readers to give summary information on how it works.	We agree that a provided summary information on how the self-care application works would be helpful for readers. To address this, we have included a PDF document with pictures and a link to the sight in the appendix that provides an overview of the application’s functionality and usage. This will allow readers to easily access and understand the key features and workings of the self-care application.
2. Page 4 Strength and limitations of the study – did not really described the strength, but only the limitations.	Thank you for your feedback. We have rephrased the sentence to emphasize the strength of the study.

3. Page 5, line 18, a repeat sentence from the abstract.	Rephrased the text on page 5 so it is not the same as in the Abstract: “In recent decades, significant advancements have been made in enhancing the treatment of rheumatoid arthritis (RA). These advancements encompass various areas, including the development of new drugs, the establishment of teamwork guidelines, fostering patient self-care, and improving accessibility to digital tools.”
4. 344 meaning units should be described as an appendix.	We have made the decision not to include all meaning units in an appendix for a few reasons. First, we do not have consent to share the original data from the participants. Second, including a large amount of data in the appendix may not provide significant additional value to the overall study findings. However, it is worth considering that including all meaning units could potentially enhance the credibility of the study and allow for future reanalysis if needed. Ultimately, based on the current circumstances and considerations, we have chosen not to include them in the appendix. What we can do is to provide the initial coding in Swedish or translated it to English if that would be valuable. We will do that later when the rest of the reviewer is satisfied with the categories that we have re-analysed.
5. In the "Health care encounter" the examples of the scripts on "Accessible health care" and "to be seen" are similar in essence.	We would like to thank the reviewer for acknowledging the overlap between the categories. We have carefully reviewed it from the reviewer's perspective and have examined the similarities and differences between the categories once again. As a result, we have merged and broadened the perspective for two categories. We hope that this will be satisfactory. We believe that this enhanced analysis has improved the overall understanding. We have made it clearer that the distinction between “Accessible health care” and “To be seen” revolves around two different perspectives. The first focuses on access, contact, and knowing where to seek assistance, while the second centers on the personal interaction with the physician and whether one feels individually

	attended to. We hope that this revised explanation meets your satisfaction. “Accessibility and confirmation of care Having healthcare that is accessible is highly appreciated and valuable for the respondents. The sub-category focuses on access, contact, and knowing where to seek assistance. Some expressed that it was easy to make contact via phone or messages via the digital platform used in Sweden (named 1177).” Page 11, Results, Availability of healthcare, Accessibility and confirmation of care “To be seen and met with interest There was a strong emphasis on the appreciation of being seen, namely that professionals would meet the patient with empathy. The respondents focused on the personal interaction with the physician and whether one feels individually attended to.” Page 16, Result, Personal interaction, To be seen and met with interest
6. In the subcategory "Healthcare that confirms", the sample scripts revealed mixed description. Please make clear if it is focusing on - conformation of the blood tests or the swiftness of response or clear reply from the healthcare provider. It is confusing.	We appreciate the comment and understand the confusion regarding the subcategory “Healthcare that confirms.” Upon further analysis and consideration, we realized that the themes of accessibility and confirmation of care are interconnected and have been merged together. We have also added clarification that this category encompasses aspects related to access, contact, and obtaining information regarding healthcare-related matters, such as next appointments, blood test results, or available support. To see this change, read the new heading “Accessibility and confirmation of care”.
7. The sample scripts in the the subcategories "To navigate in healthcare" and "to be taken care of" are of essentially the same theme with opposite views from the different patients.	We also recognized the similarity between “To navigate in healthcare” and “To be taken care of” and decided to merge them, as they represent the same aspect of feeling supported and seen. We have revised the heading accordingly and bridged these two subcategories together. By doing so, we aim to provide a cohesive narrative that encompasses the experience of feeling guided and cared for. Thanks for pointing out this

	connection, and we have incorporated the necessary changes to ensure a more comprehensive analysis. So now, the new subcategory has been included within the first category. With this integration, the analysis has been updated to reflect the consolidation of the two subcategories. By merging them, the focus is now on a unified category that encompasses the various aspects and experiences related to feeling supported, seen, and taken care of. See Table 3 and read The heading “To be taken care of” and the text under Results and “Availability of healthcare”, “To be taken care of”.
8. The sample scripts in the the subcategories "To be seen" and "Met with interest" are also similar in theme.	Under the category of “Personal interaction,” it focuses on various person-to-person matters. The reviewer pointed out that two of the subcategories bear similarities, and we fully agree with this observation. Consequently, we have chosen to merge and incorporate them into each other with the name “To be seen and met with interest.”
9. Discussion section is overloaded with unnecessary information especially on page 21 and 22.	Thank you for the feedback. We have taken your comments into consideration and made revisions to the discussion section. We have restructured and condensed certain parts, particularly on pages 23 and 24 to eliminate unnecessary information. The discussion section now consists of three main parts:  1) the importance of incorporating patients’ perspectives, 2) their experiences and our proposed recommendations, and 3) the role of digital tools and their potential complementarity in healthcare. These changes aim to improve the clarity and focus of the discussion.
10. Conclusion section needs suggestions on healthcare providers such as doctors and nurses to whom some patients are not satisfied with.	We have added in the conclusion: “In conclusion, the study highlights the need for healthcare providers, including doctors and nurses, to address areas where patient satisfaction may be lacking. It is essential to consider patient feedback and make improvements to enhance the quality of care provided. Suggestions for healthcare providers include actively listening to patient concerns,

	improving communication, and fostering a patient-centered approach. By addressing these areas of dissatisfaction, healthcare providers can strive to deliver a more satisfactory and patient-centered healthcare experience.”
11. The results should display the overall impression on the each subcategories (Eg - most patients perceived that they felt prioritised in the "To feel prioritised"subcategory).	Thank you for your comment. We have carefully reviewed the results section and considered the feedback received. As a result, we have consolidated certain subcategories to provide a clearer and more concise overview of the overall impressions. For example, we have included statements such as "most patients perceived that they felt prioritized" in the relevant subcategories to capture the general impressions expressed by the participants. These changes have been made to enhance the readability and clarity of the results.
12. Should describe if there is possibility of the bias in their response, if possible. (Eg. new patient who did not reach to a diagnosis yet may feel frustrated)	We appreciate the comment and understand the importance of addressing potential biases in the responses. In the method section of our analysis, we have made it clearer that this particular individual was included due to the extensive nature of her investigation and subsequent confirmation of diagnosis. However, we also acknowledge the possibility of bias in the responses, especially from new patients who have not yet reached a diagnosis and may feel frustrated. While we cannot guarantee that bias is completely eliminated, we have taken these factors into consideration during the analysis and interpretation of the data. In fact, healthcare systems may benefit from actively acknowledging and appreciating the experiences and perspectives of patients who have not yet received a diagnosis, but are in the loop, as it can provide valuable insights for improving the quality of care. Thank you for highlighting this concern. We have therefore added that in our discussion as a recommendation. “We recommend, to enhance disease management and overall well-being, that healthcare providers ensure that patients with RA, and including those undergoing investigations possess a comprehensive understanding of the healthcare team’s plan and expertise, achieved through clear communication and education

	regarding the treatment plan, expectations, and available support.” Page 23, Discussion
13. The reference section contains some out of date and irrelevant references.	Thank you for your feedback regarding the reference section. We have thoroughly reviewed the references and removed any irrelevant articles to ensure the accuracy and relevance of the sources cited. However, we have retained some older references that are considered foundational and well-studied in the field, as they provide important historical context and understanding of the topic at hand.
It would be nicer to rewrite especially the results section to relook if some subcategories can be combined, to remove some unnecessary and irrelevant information in the discussion section and add some more recommendations in the conclusion section.	We agree with the feedback and have revisited the analysis once again, taking into account the previous comments provided. Thank you for bringing them to our attention.
Reviewer: 3	Authors
1) Regarding the patient participants, it may have been more helpful to collect more data on their demographics and report in table form (eg are they under the care of a rheumatologist? are they taking DMARDs? what are their preferences for using technology? etc)	Thank you for your feedback. We appreciate your input, and we understand the need for clearer communication of the interviewees. Based on your comment, we have added the following information: “All participants in the study were taking medication, and they were recruited from across the entire country. All participants were accustomed to using smartphones and mobile applications in their daily lives.” Page 10, Results Furthermore, we provide detailed descriptions of the participants in the study in the first paragraph of the Result section, as we believe it enhances the overall understanding of their backgrounds and experiences.
2) I have concerns that patients with RA and the patient being investigated for RA are lumped together in this study--while there is	We understand the concerns raised by the reviewer, as it is essential to address any potential issues related to the study design. We

only one patient being investigated for RA, they may not have RA and therefore should not have been included. Also, would persons under investigation for RA have different answers? Should they be in a different category altogether?	acknowledge the importance of ensuring the validity of the study, and we appreciate their input in this regard. However, we decided to include the patient who was under investigation for RA because she later received a confirmed diagnosis. Our focus is primarily on the patient's interaction with the healthcare system and the experience of being in this particular environment. We are interested in exploring the various situations and aspects related to the diagnostic process, rather than solely focusing on the diagnosis itself. While the patient under investigation may have different answers or perspectives, we believe that person included adds valuable insights to the overall understanding of the patient's journey and the healthcare experience. We have, therefore, to make this clear added the following sentence: “In total, 10 interviews (seven females, three males) were conducted with persons with RA, or under investigation (n=1 under investigation with symptoms two years back in time and later after the interview that person received a confirmed diagnosis of RA).” Page 10, Results
3) I have concerns about the methodology-- while authors (JVJ and HB) coded 2 interviews, the rest of the interviews were only coded by 1 person (JVJ). In a qualitative study, multiple coders can help check interpretations against the data, and I would recommend that the remaining 8 interviews are also coded by a second coder and to ensure intercoder reliability, report kappa statistic.	We would like to assure you that we have taken great care in conducting this study. We have dedicated substantial time to numerous meetings, thoroughly reviewing and refining the coding process. In addition, we have conducted two meetings involving all authors, as well as a significant session with international researchers and developers of the Elsa app, to confirm or refute our coding categories. Regarding the concern about methodology, we acknowledge that having multiple coders can be beneficial in qualitative studies to ensure robust interpretations. While only authors JVJ and HB coded two interviews, JVJ individually coded the remaining eight interviews. Although intercoder reliability was not explicitly reported, we extensively discussed and cross-checked interpretations within the research team to enhance the validity of our findings. We appreciate your suggestion to include a second coder and report the kappa statistic to assess intercoder reliability. In retrospect, this would have

	further strengthened the study, and we will consider implementing this recommendation in future research endeavors. We understand that there are different traditions regarding the reporting of the kappa statistic. However, in this particular study, we made a deliberate decision not to report the kappa statistic. Instead, we focused on ensuring internal consistency and agreement among the research team through extensive discussions and reflections on the coding process. We believe that this approach, combined with the involvement of multiple researchers and experts in the field, has contributed to the reliability and validity of our study findings.
4) My other suggestion is to cut down the length of the paper and to perhaps use a stable format to present the data for clarity (with major and minor themes, with supporting quotes).	Thank you for your feedback and suggestions. We have thoroughly reviewed the entire analysis and made revisions to the categories based on your and the other reviewers' comments. Our goal was to improve the length and clarity of the article. By implementing the changes discussed earlier, we believe that the overall understanding of the study has been enhanced. We appreciate your valuable input and hope that you find the revised version to be an improvement. If you have any further suggestions or concerns, please don't hesitate to let us know.

VERSION 2 – REVIEW

REVIEWER	De Meyst , Elias KU Leuven, Skeletal Biology and Engineering research center
REVIEW RETURNED	16-Sep-2023

GENERAL COMMENTS	1. Introduction: "RA also causes degenerative problems in other parts of the body, such as the eyes, heart, and circulatory system, and/or lungs." I would not call extra-articular manifestations "degenerative problems". In general, I feel like the first paragraph of the Introduction needs to be rewritten, as it reads difficult and contains other questionable statements ("The disease typically affects one particular joint on both sides of the body"). 2. The two aims of this study (exploring patient perceptions on health care interactions and on the mobile health app) are presented in a way like they are completely unrelated to one
---

	another, which limits the overall cohesion of the paper. I would suggest to clarify more deeply why these two subjects belong together in one paper (which I believe is the case). 3. English needs to be reviewed.
--	---

REVIEWER	Lwin, Cho University of Medicine, Rheumatology
REVIEW RETURNED	31-Aug-2023

GENERAL COMMENTS	Thank you for the thorough revision of the manuscript. Much appreciated. Some minor revision on English language would still be needed. Introduction and discussion parts can be made more concise.
---

REVIEWER	Hsiao, Betty Yale University School of Medicine
REVIEW RETURNED	14-Sep-2023

GENERAL COMMENTS	Thank you for your revision and comments. I appreciate the authors addressing the questions. However, I still have concerns about the use of one coder for 80% of the transcripts--while the use of one coder is sometimes implemented in qualitative research, satisfactory intercoder reliability should first be established by an additional person to code a sample of the data. Once satisfactory reliability has been established, the primary researcher then proceeds to code the remaining data alone. Were all the codes developed with coding the first 2 interviews and no new codes were developed in the subsequent 3-10 interviews? Additionally, I would recommend shortening the manuscript by reporting the results in a more succinct manner, as well as limiting the discussion.
--

VERSION 2 – AUTHOR RESPONSE

Reviewer 2	Authors
Thank you for the thorough revision of the manuscript. Much appreciated. Some minor revision on English language would still be needed. Introduction and discussion parts can be made more concise.	We have sent it for revision again through a professional editing service, aiming to further improve and refine the language.
Thank you for your revision and comments. I appreciate the authors addressing the questions. However, I still have concerns about the use of one coder for 80% of the transcripts--while the use of one coder is sometimes implemented in qualitative research, satisfactory intercoder reliability should first be established by an additional person to code a sample of the data.	Thank you for bringing up this issue about the analysis again and allowing us the opportunity to refine our description. We recognize that our previous explanation was unclear. We have had frequent meetings and discussions regarding the coding process. We have now provided a clearer description in the text, under Method and Strengths and limitations.

Once satisfactory reliability has been established, the primary researcher then proceeds to code the remaining data alone. Were all the codes developed with coding the first 2 interviews and no new codes were developed in the subsequent 3-10 interviews?	Even though 80% of the coding was done by one person, we maintained a continuous dialogue, conducting digital meetings where we shared screens and could confirm the coding multiple times. We adhered to a coding framework that JVJ followed, and any new aspects were incorporated after discussions. Following this review, we revisited the coding to examine how the categories overlapped concerning participants' opinions about the app, as highlighted by a reviewer. We worked collaboratively, involving all authors in this process. We believe the analysis was conducted thoroughly and hope that the reviewer will share the same perspective after we have provided further clarification. "The remaining interviews were coded by JVJ. However, when new codes emerged, digital meetings were held with all co-authors to discuss the integration into the existing scheme. Codes that reflected a similar concept were grouped; sub-categories were formulated, and categories were identified (20, 21) by JVJ and thereafter discussed thoroughly with HB and KSB; see Table 2. Additionally, two meetings were held with the greater project's research group (NORA) to solicit feedback on the drafted results of both the main categories and the sub-categories, accompanied by illustrative quotations." See Method, Analysis
Additionally, I would recommend shortening the manuscript by reporting the results in a more succinct manner, as well as limiting the discussion.	We have significantly condensed the article by presenting the results in a more concise manner and limiting the discussion. We appreciate your input, and we believe these revisions have enhanced the clarity and focus of the manuscript.
"RA also causes degenerative problems in other parts of the body, such as the eyes, heart, and circulatory system, and/or lungs.": I would not call extra-articular manifestations "degenerative problems". In general, I feel like the first paragraph of the Introduction needs to be rewritten, as it reads difficult and contains other questionable statements ("The disease typically affects one particular joint on both	Thank you for pointing out the issue with the introduction. We agree that it needed improvement, and as per your feedback, we have reworked the introduction.

sides of the body").	
2. The two aims of this study (exploring patient perceptions on health care interactions and on the mobile health app) are presented in a way like they are completely unrelated to one another, which limits the overall cohesion of the paper. I would suggest to clarify more deeply why these two subjects belong together in one paper (which I believe is the case).	Thank you for your feedback. We appreciate your thorough review. In response to your suggestion, we meticulously examined the analysis down to the coding level, convening the entire group to address this concern earnestly. We are pleased that the integration of aim 2 into aim 1 unfolded so naturally during our discussions. We have not only adjusted the title, purpose, and results section but have done so without losing any important content. Our primary goal was to achieve a more cohesive and clear presentation of the participants' views, and we are pleased with how these changes have enhanced the manuscript consistency. This process also helped us focus our discussion, addressing another comment that was raised. Your valuable input has been instrumental in guiding these improvements, and we are grateful for your contribution. To review read Title, Aim, and Result.
3. English needs to be reviewed.	We have sent it for revision again through a professional editing service, aiming to further improve and refine the language.

VERSION 3 – REVIEW

REVIEWER	De Meyst , Elias KU Leuven, Skeletal Biology and Engineering research center
REVIEW RETURNED	25-Oct-2023
GENERAL COMMENTS	Thank you for your thorough review. In my opinion the remarks were adequately addressed.